# TeV Instrumentation: Current and Future

Julian Sitarek 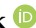

Institute for Cosmic Ray Research (ICRR), The University of Tokyo, Kashiwa 277-8582, Japan; jsitarek@uni.lodz.pl

**Abstract:** During the last 20 years, TeV astronomy has turned from a fledgling field, with only a handful of sources, into a fully-developed astronomy discipline, broadening our knowledge on a variety of types of TeV gamma-ray sources. This progress has been mainly achieved due to the currently operating instruments: imaging atmospheric Cherenkov telescopes, surface arrays and water Cherenkov detectors. Moreover, we are at the brink of a next generation of instruments, with a considerable leap in performance parameters. This review summarizes the current status of the TeV astronomy instrumentation, mainly focusing on the comparison of the different types of instruments and analysis challenges, as well as providing an outlook into the future installations. The capabilities and limitations of different techniques of observations of TeV gamma rays are discussed, as well as synergies to other bands and messengers.

**Keywords:** gamma rays; Cherenkov telescopes; water Cherenkov detectors; surface arrays; analysis methods

## 1. Introduction

Out of the different messengers (see Chapter 7 of this Special Issue) very-high-energy (VHE, $\gtrsim$100 GeV) gamma rays so far have been the most successful in investigating the most extreme, high-energy processes in extragalactic sources. Cosmic rays, except for those exceeding EeV energies, are isotropized by Galactic and extragalactic magnetic fields and hence cannot be pinpointed to individual sources. Neutrino astronomy, although being able to unambiguously point out the occurrence of hadronic processes in the source, is severely marred by the very small statistics of detectable events. Gamma-ray astronomy solves both of those problems, as high-energy photons can be easily detected by their interactions with matter, and cosmic gamma-ray sources produce emissions associated with a particular direction in the sky.

Due to the interaction of gamma rays with the Earth's atmosphere, they cannot be observed directly using ground-based instruments. Instead, balloon-based (see, e.g., GRAINE [1]), or particularly successful space-borne (see e.g., the large area telescope on board the *Fermi* satellite, [2]) instruments are effective in monitoring the sky at GeV energies. Nevertheless, at the VHE range, such a technique is no longer efficient. As balloon and satellite experiments have collection areas comparable to their physical size (of the order of m$^2$), even for bright sources this allows detection only of the order of a single event per day above 100 GeV. Such low event rates prevents investigations of short-term phenomena in this energy range through the direct detection of gamma rays.

Thankfully, the absorption of gamma rays in the atmosphere can be turned into an indirect method of detecting radiation in the VHE range. A primary gamma ray is converted into a $e^+e^-$ pair in the radiation fields of atmospheric nuclei. These in turn produce further gamma rays through the Bremsstrahlung process. The combination of both processes results in an electromagnetic cascade, propagating through the atmosphere. The cascade, often referred to as an (extensive) atmospheric shower, initiated by a TeV gamma ray, is composed of thousands of highly relativistic particles. Such events can be observed directly by detecting the particles surviving down to the ground level using surface arrays (SAs) or water Cherenkov detectors (WCDs). Alternatively, the events can be studied through

observations of Cherenkov light emitted by particles during the shower development (imaging atmospheric Cherenkov telescope (IACT) techniques). Instruments exploiting both of those techniques have allowed us to expand the catalogue of VHE sources up to ∼250 objects (see Figure 1).

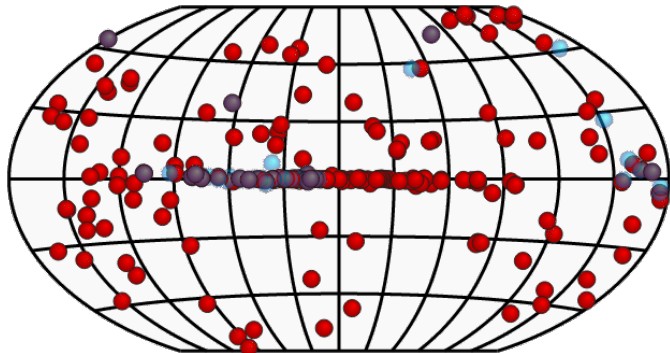

**Figure 1.** Overlay image in Galactic coordinates of the sources seen via the currently operating major IACT experiments (H.E.S.S., MAGIC, VERITAS, denoted in red) and via currently operating WCD and SA experiments (HAWC, LHAASO, in blue). Sources detected using both techniques are indicated in violet. Data obtained using the TeVCAT (http://tevcat2.uchicago.edu accessed on 18 October 2021) service.

Although the IACT technique has been used to detect both Galactic and extragalactic sources, so far most of the SA/WCD sources have been claimed in the Galactic plane. In this review, both techniques are discussed, along with their advantages/disadvantages and complementarity. The current and future instruments are reviewed, as well as their connections to other bands and messengers.

## 2. Instrumentation

Both techniques share the same principle of reconstructing the primary gamma ray events based on the properties of their extensive air showers (see Figure 2).

They also struggle with the same issue of a dominant background due to similar showers produced by isotropic cosmic ray particles. Nevertheless, the usage of either the Cherenkov light to track the development of the shower or the remaining shower particles reaching the ground causes a number of differences between the two approaches, strongly affecting the performance achieved.

### 2.1. IACT Technique

A shower produced by a TeV primary gamma ray contains thousands of ultra-relativistic particles. Charged particles moving faster than the light in the atmosphere will induce the production of dim and short (of the order of nanoseconds) flashes of Cherenkov light. As the TeV showers reach their maximum at the height of ∼10 km a.s.l., and the Cherenkov radiation is emitted at an angle of ∼1°, the radiation is spread over a region with a radius of ∼120 m, the so-called light pool. A telescope located in such a light pool can detect the Cherenkov emission. Hence, despite the fact that the physical sizes of the current-generation IACT telescopes are few hundred $m^2$, they can achieve effective collection areas for TeV gamma rays, of the order of $10^5$ $m^2$.

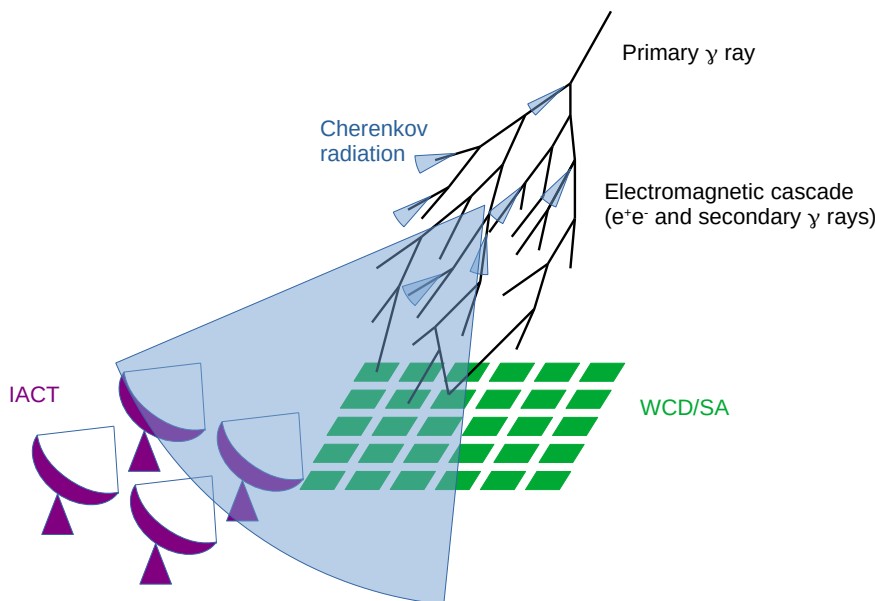

**Figure 2.** Principle of TeV gamma ray detection. Primary gamma ray interacts with atmospheric nuclei, creating a cascade (atmospheric shower) of secondary particles (black lines). Charged particles, with their speed exceeding the speed of light in the atmosphere, induce the emission of Cherenkov light flashes (blue cones). The radiation can be observed using IACTs (violet). Those of the secondary particles that reach the ground level can also be detected directly through WCD or SA experiments (green).

The two main components of IACTs are their mirror dish (including a drive system to reposition it) that collects the Cherenkov photons and the camera (as well as the trigger and data acquisition systems) that allows the conversion of Cherenkov light into an image of the shower. The size of the mirror dish determines the energy threshold of an IACT. The smaller ($\lesssim$15 m diameter) telescopes employ the Davies–Cotton shape of the mirror dish, which minimizes optical aberrations. For larger telescopes ($\gtrsim$15 m diameter), a parabolic shape is used, since it minimizes the time spread of isochronous radiation reaching the telescope. Due to the large sizes of the telescope dishes, tessellated mirrors are used. Unless the telescope structure is stiff enough [3], the changing gravitational loads as the telescope moves require constant correction of the individual direction (see [4] and references therein) to maintain accurate pointing.

One of the key features that added to the success of IACTs is the "imaging" capabilities of their cameras. The light collected by the telescope is focused on an optical plane composed of the order of 1000 photosensors (most commonly photomultipliers (PMTs); however, silicon photomultipliers (SiPMs) are also being considered or already used, in [5]). The camera allows one to construct the two-dimensional angular distribution of the observed light, providing the image of the shower. Such images can be parametrized as ellipses, which facilitates the reconstruction of the properties of the primary particle [6]. Thanks to the changing atmospheric refractive index with the height a.s.l., the long axis of the ellipse tracks the longitudinal evolution of the shower. On the other hand, the short axis is related to the latitudinal distribution. The angular image from a single telescope is, however, not sufficient to fully distinguish between showers that reach their maximum at the same line of sight, but at different heights above the ground (or alternatively at different impact distances). The combination of images of the same event seen by multiple nearby telescopes provides a natural way to reconstruct the three-dimensional geometry of the event (see e.g., [7]). A stereoscopic trigger is also an efficient way of removing the background of single-muon events both at the hardware level (see, e.g., [8]) and during the analysis (see, e.g., [9]). The three major, currently operating, stereoscopic IACT systems are H.E.S.S. [10], MAGIC [11] and VERITAS [8].

### 2.2. SA and WCD Techniques

In contrast to IACTs, surface detectors directly exploit the particles that reach the ground level. One way of detecting those particles is through the use of surface arrays (SA[1]) and scintillation counters (see, e.g., [12,13]), that track passing of charged particles. A small layer of dense material (typically lead) placed above such a counter provides a means of the conversion of secondary gamma rays into $e^+e^-$ pairs (as well as slightly raising the detection energy threshold for the secondary charged particles). The net effect of such an absorption layer is an improvement of the energy and angular resolutions and a decrease of the energy threshold for primary particles.

Although SAs are usually finely spaced to maximize the collection area at the highest energy, it is also possible to construct dense or full-coverage arrays. The ARGO-YBJ experiment used resistive plate chambers to cover a dense carpet of $72\,\mathrm{m} \times 76\,\mathrm{m}$ with 92% coverage, and an additional guard ring, with a total area of $110\,\mathrm{m} \times 100\,\mathrm{m}$ [14].

Alternatively, WCDs use water tanks to absorb and detect the secondary particles produced in the shower. Secondary charged particles moving faster than about $0.77\,c$ will induce the production of Cherenkov radiation inside such a tank, which can be gathered using large-sized PMTs (typically a photocathode with a diameter of over $20\,\mathrm{cm}$). Although SA stations basically detect the number of charged particles (rather than energy deposits), it is more complicated in the case of WCDs. Electrons/positrons reaching a WCD detector encounter multiple radiation lengths of water that efficiently absorb them, facilitating a calorimetric measurement of this component. However, muons pass through the detector, producing a signal that is not proportional to their energy, but rather the detector crossing length.

The previous generation of WCDs, such as MILAGRO [15], used a single pond of water and a grid of PMTs. In such a setup, the scattered Cherenkov radiation could travel to neighbouring PMTs, blurring the response of the detector. In the HAWC [16] and LHAASO-WCDA [17] system, separate tanks or curtains were used to counteract such an effect. At a sufficiently high level above the energy threshold, the collection area of the gamma rays is closely related with the physical size of the array. On the other hand, the energy threshold of the instrument is determined by the height a.s.l. of the WCD and its particle collection efficiency (which depends also on the fill factor of the array, i.e., how large a fraction of the total array area is covered by the detectors). In order to limit the rate of random coincidences, a trigger can exploit the time correlation of signals of different parts of detectors, depending on their relative distance (see [18] for details of the ARGO-YBJ).

The content of the shower particles depends on the primary particle type. Although gamma-ray initiated showers produce nearly exclusively gamma-ray and $e^{\pm}$ particles, in the case of hadron-induced showers, there is an additional component represented by muons, which are much more penetrative. Therefore, the inclusion of additional particle counters located under a natural, deep layer of absorbing material (typically earth or water) provides a measurement of the muon content of the shower, which can be used in the selection of gamma-ray showers [19].

The sampling of local (and time-dependent) secondary particle distributions on the ground by means of WCD or SA detectors is, in a sense, equivalent to the imaging capabilities of IACTs. Both methods construct an image of the shower (in the ground coordinates for SAs/WCDs tracking the lateral distribution of particles and in angular coordinates for IACTs, also tracking the longitudinal profile of the shower) and through this image, allow us to determine the structure of the atmospheric shower, and thus reconstruct the energy and arrival direction and also reduce the background interference from cosmic-ray-induced showers.

### 2.3. Comparison of Performance and Synergies

In Table 1 the basic performance parameters are summarized and compared for IACT and WCD/SA detectors.

**Table 1.** Comparison of the typical performance parameters of IACT and SA/WCD instruments. Due to the energy dependence of the performance parameters, the energy resolution, angular resolution and sensitivity are given for the best sensitivity range of IACT or SA/WCD instruments (i.e., about an order of magnitude higher than their corresponding energy thresholds). The values for individual instruments of a given type can vary by a factor of a few, depending on the details of the hardware implementation and analysis methods.

| Characteristic | IACT | SA/WCD |
|---|---|---|
| Energy threshold | $\sim$ tens of GeV (for a few hundred m$^2$ mirror dish) | $\sim$TeV |
| Duty cycle | $\sim$ 10% | $\lesssim$ 100% |
| Field of view | $\sim$ a few millisr | $\sim$ sr |
| Energy resolution | $\sim$ 15% | $\sim$ 40% |
| Angular resolution | $\sim 0.1°$ | $\sim 0.2°$ |
| Sensitivity | $\sim$ 1% Crab Nebula flux in 25 h | a few % Crab Nebula flux in 5 yr |
| Main present instruments | H.E.S.S., MAGIC, VERITAS | Tibet AS-$\gamma$, HAWC, LHAASO-WCDA, LHAASO-KM2A |
| Future instruments | CTA | SWGO, ALPACA |

The energy dependence of the angular and the energy resolution for IACT and WCD instruments is shown in Figure 3. The performance parameters of both types of instruments in general improve with energy, however at the highest energies they might again worsen due to saturation effects. The maximum energy up to which the source can be studies is strongly dependent on both the geometrical footprint of the instrument (favouring sparse arrays), but also on the spectral shape and flux of the observed source.

IACTs detect the shower development in the atmosphere, rather than the tail of the shower that reaches the ground. This has a profound impact on the energy threshold and the low-energy (sub-TeV) performance of those instruments. Such low-energy showers will develop mainly in the atmosphere, providing ample information, and only a small number of particles will reach the ground, providing little input for the WCD and SA techniques. Sparse SAs are sensitive only above about 10 TeV. The full-coverage approach used in some WCDs and, e.g., ARGO-YBJ allows observations starting from (sub-)TeV levels. On the other hand, WCD and SA instruments are not limited to observations during (preferentially moonless) nights, and in addition have a large field of view (FOV), and can thus continuously monitor a large fraction of the sky. This also leads to difficulty in the comparison of the sensitivity of the two types of instruments. Although for transient phenomena and short-term variability IACTs are clearly superior, the possibility of integration of a large exposure from a WCD or SA instrument from its broad FOV provides an efficient method both for studies of steady sources, and for unbiased studies of the typical states of variable emitters. Those differences are reflected in the exposure times quoted for sensitivity calculations. In the case of WCDs and SAs, the sensitivities are typically quoted for 1 or 5 years, and they are related to the total, aggregated time of instrument operation. On the contrary, for IACTs, 50 h are typically used as a reference number. Although it is theoretically possible to observe a single source with an IACT even for 300 h within a single season, in practice this is usually not performed as the sensitivity in (sub-)TeV range improves only with the square root of the time, and extensive exposures would block the observation time of other targets. In addition, at the lowest energies the sensitivity starts to be limited by the systematic uncertainties on the background, preventing further improvement with increasing observation time.

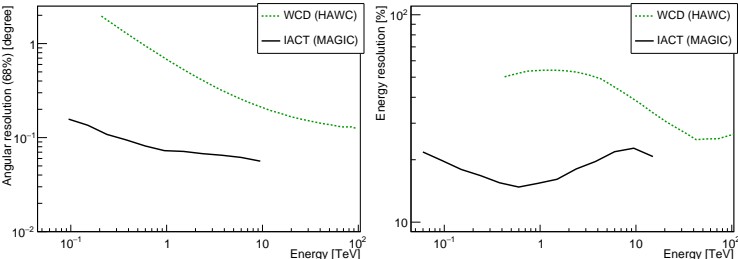

**Figure 3.** Comparison of the angular (left panel) and energy (right panel) resolution of a WCD (HAWC, dotted green line) and an IACT (MAGIC, using look-up table energy estimations, solid black line). HAWC angular resolution is taken from https://www.hawc-observatory.org/observatory/sensi.php (accessed on 19 November 2021), and energy resolution (for neural networks analysis, recalculated from $\log_{10}$ space to linear space) from [20]. MAGIC performance in both panels is taken from [21].

As IACTs probe a significant fraction of the shower developments (in particular around the shower maximum) via an imaging technique, they are fairly good at calorimetry of the parent particle. The achieved energy resolution of IACTs in general improves with energy, and depends strongly on the energy estimation method; however, values of the order of 15% are typical. On the other hand, the number and total energy of particles in the tail of the shower that are accessible to the WCD and SA techniques is subjected to high shower-to-shower fluctuations. This effect strongly degrades the energy resolution of the WCD and SA instruments.

IACT and WCD instruments both achieve angular resolutions of the order of 0.1° in their highest energy ranges. However, as the energy range is different for both types of instruments at low (TeV and sub-TeV) energies, IACTs clearly provide higher performance. At energies of tens of TeV, both types of instruments have excellent angular resolution. However, WCDs/SAs excel due to their higher duty cycle, which improves their sensitivity and thus facilitates the investigation of source morphology.

Because of the greatly different duty cycles of both types of instruments, and their instantaneous performance, it is difficult to compare them fairly in terms of sensitivity. The same sensitivity of approximately 3% of Crab Nebula flux in a given sky location is achieved by WCDs in about 5 years and by an IACT in just ∼3 h. However, the WCD would simultaneously scan about half of the sky with such a sensitivity. To cover the same fraction of the sky with such a sensitivity, IACTs would require about 8000 h of observation time (assuming 2 msr FoV), which corresponds to 8 years of observations with the typical duty cycle of IACTs. Therefore, large-scale unbiased sky surveys with IACTs are feasible only for regions in which an enhanced number of sources is expected, such as the Galactic plane [22] or the Cygnus region [23]. On the other hand, WCDs have turned out to be excellent survey instruments, in particular for the highest-energy Galactic sources [16,24].

Large-FOV instruments can also serve as the source of a trigger for more detailed observations with high-sensitivity, pointed instruments. This synergy between WCDs/SAs and IACTs can be used either to study stable sources discovered with deep WCD/SA observations or to follow up on short-term flares. VERITAS and MAGIC telescopes have performed follow-up observations of sources discovered in the HAWC catalogs [25,26]. Surprisingly, those studies did not confirm TeV emissions from most of the reported sources. A likely explanation for this apparent tension between the results of both techniques is given by the possible 1° scale extension of these sources. At the energy of ∼TeV, such an extension is comparable to the angular resolution of HAWC (and thus would also not affect its sensitivity strongly), whereas it would degrade the sensitivity of IACT observations by about an order of magnitude, and in more extreme causes could cause confusion of the source emission with the isotropic background. In fact, a joint study of H.E.S.S. and HAWC was performed, in which IACT data were artificially smeared to match the WCD angular resolution and a similar method for background subtraction was used for both types of instruments. It showed that both techniques provide a consistent view of the Galactic

plane, with the remaining differences in the gamma-ray flux explainable by systematic uncertainties [27].

The second possibility, the triggering of IACTs based on short-term flares seen by WCDs/SAs, has been much less explored to date. The main reason for this is the short-term sensitivity and limitations due to the false alarm rate over a broad FoV. This makes detection efficient only for very strong flares, larger than a few times the flux of the Crab Nebula [28], which have been observed only in a handful of the brightest blazars.

### 2.4. Hybrid Arrays

The synergy of both types of techniques allows one to combine them within the same facility. This has been accomplished already in the previous generation of instruments with the combination of HEGRA IACTs, the wide-acceptance Cherenkov array AIROBICC and surface detectors (scintillator arrays) [29].

A modern version of such a combination is LHAASO, a hybrid array combining various TeV observations techniques to achieve excellent performance over a broad energy range both for observations of gamma rays and cosmic rays [30]. The facility was completed in 2021 and it is starting to provide its first scientific results (see, e.g., [24]). It comprises, among others, LHAASO-KM2A—a sparse array of electromagnetic surface detectors (scintilation counters) and an underground WCD for muon detection; LHAASO-WCDA (a water Cherenkov detector array)—with three tessellated water ponds; and LHAASO-WFCTA (wide-field air Cherenkov telescope array)—an array of 18 IACTs with a wide FOV ($16° \times 16°$).

### 3. Future Instruments

In the coming years, the next generation of TeV instruments will be constructed. In the case of IACT instruments, a major leap in performance is expected between the current generation and the Cherenkov telescope array (CTA, [31,32]). The CTA will be composed of about one hundred telescopes in three different sizes (LST, MST and SST). In order to provide all-sky coverage, the telescopes will be distributed in two locations (one in the Northern and one in the Southern Hemisphere). The arrays of four LSTs (large-sized telescopes), due to their 23-m diameter reflectors, will allow one to firmly extend the IACT technique into the still-poorly-exploited region of tens of GeV. SSTs (small-sized telescopes) are large (70 units) arrays of telescopes of a few meters in diameter. They are aimed at providing optimal sensitivity, angular resolution and surveying capabilities at energies of tens of TeV [33], making them competitive instruments with WCD/SA techniques. The necessity of large FOVs for SSTs, without introducing large optical aberration, stimulates the usage of previously unexplored or weakly explored technologies for IACTs, such as dual-mirror (Schwarzschild-Couder) designs and the use of SiPM light detectors. MSTs (medium-sized telescopes) are the most similar in diameter to the current generation of IACTs. Nevertheless, their large number (15 and 25 in Northern and Southern CTA sites) will allow improvements of their sensitivity in the canonical energy range of IACTs (~TeV) by an order of magnitude, and their large FOV will allow one to perform sensitive scans of large parts of the sky. Two different designs are being pursued for MSTs: in addition to the classical single-mirror design, a dual-mirror design is also being developed [34].

The sensitivity of WCD instruments is mainly in the TeV range. In the case of distant extragalactic sources, the TeV emission is severely absorbed in the pair production process in extragalactic background light [35]. The capability of studying sources with a redshift $\gtrsim 0.1$ with WCD instruments is therefore strongly limited [36]. On the other hand, Galactic sources often show broadband spectra extending into tens of TeV (and beyond). Thus, it is unfortunate that all currently operating WCD and SA installations are located in the Northern hemisphere (Tibet AS-$\gamma$ at 30° N, HAWC at 19° N, LHAASO at 29° N), from which location most of the Galactic plane is not observable. The Southern Wide-field Gamma-ray Observatory (SWGO, [37]) is a project aiming to build a WCD experiment in

the Southern hemisphere. The project is currently at an early design phase, with various detector concepts [38].

Another planned project that will exploit a surface array of detectors for gamma-ray observations in the Southern hemisphere is ALPACA and its prototype array ALPAQUITA [39]. ALPACA will have a similar design to the Tibet AS$\gamma$ experiment, with a sparse array of scintillation detectors and underground muon detectors.

A comparison of the sensitivity of various presently operating and future instruments is shown in Figure 4.

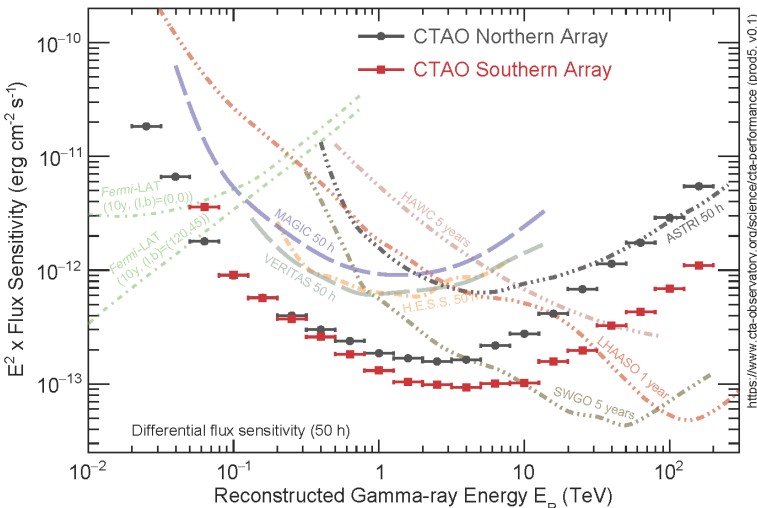

**Figure 4.** Comparison of the sensitivity of present and future GeV and TeV instrumentations: ASTRI [40], CTA [41], *Fermi*-LAT (http://www.slac.stanford.edu/exp/glast/groups/canda/lat_Performance.htm, scaled), HAWC ([42], scaled), H.E.S.S. (adapted from [43]), MAGIC [21], LHAASO [44], SWGO [37] and VERITAS (https://veritas.sao.arizona.edu/about-veritas/veritas-specifications). Image reproduced from https://www.cta-observatory.org/science/ctao-performance/ (accessed on 8 November 2021).

The construction of the next generation of instruments will allow us to perform studies in energy ranges that have been poorly investigated so far (tens of GeV and above tens of TeV) and significantly improve the sensitivity in TeV energies. This is expected to raise the detection of individual sources of rare classes (such as gamma-ray bursts, novae and some sub-classes of active galaxies) up to the level of population studies. Discoveries of gamma-ray emissions from classes of objects previously unknown in this energy range is also expected. Finally, the survey capabilities of those instruments will allow an unbiased search for gamma-ray emissions.

## 4. Analysis Methods

The analysis of TeV data from both IACT and WCD/SA instruments can be divided into three stages. Low-level analysis takes care of the calibration (both in terms of reconstructed signal strength and its timing) of individual photomultipliers. In medium-level analysis, the information from the individual detectors is aggregated and, based on this, the basic parameters of a single event (such as event type, its energy and arrival direction) are determined. In the case of WCDs and SAs, instead of physical shower parameters, more experimentally-oriented quantities are sometimes used in this step (e.g., instead of event energy, the fraction of tanks hit by the shower). In the final stage of analysis, the individual event information is used to derive high-level information about the source (such as the energy spectrum, morphology of the emission or time-variability).

As both WCD/SA and IACT instruments share the same principles of the reconstruction of air showers, some of their analysis problems are shared as well. In particular, gamma-ray sources are immersed in a few orders of magnitude more abundant isotropic

background noise induced by cosmic ray showers. The sensitivity of TeV instruments is mainly dependent on how well such a background can be rejected and how large systematic uncertainties are induced through the subtraction of the residual background.

### 4.1. Event Cleaning

In both techniques, low-energy showers, which are actually the most abundant in the spectra of cosmic sources, produce smaller responses in the instrument, which are more affected by noise. This results in only a small fraction of pixels (in IACTs) and tanks/detector stations (in WCDs and SAs, respectively) carrying the shower information. For such events it is essential to determine which parts of the detector are dominated by the shower rather than by the noise, in order to achieve a low energy threshold and good low-energy performance.

In the case of IACTs, the brightest pixels in the camera (above a given threshold) form the core of the image. An additional condition is used to accept pixels of lower brightness as well if they are neighbouring one of the core pixels [45]. The usage of signal timing significantly improves the reconstruction of low-energy showers [46,47].

In the case of WCDs, the events in which the number of triggering tanks is too low are excluded from the analysis, and the events are classified based on how large the fraction of all the tanks is that provides a trigger [42].

### 4.2. Event Reconstruction and Background Rejection

In the case of IACTs, the most commonly used background rejection method is still based on the classical Hillas parametrization of the events [6]. Due to the intrinsic differences of gamma-ray-induced and hadronic air showers, the latter tend to be more spread out, in particular in the lateral direction. Additionally stereoscopic observations of the same event with multiple IACTs allow the 3D reconstruction of the event, including the estimation of the height of the peak of the Cherenkov emission (which is a measure of the maximum of the shower development). This allows for an efficient rejection of single-muon events that could mimic gamma-ray showers (see, e.g., [9]) and for a partial rejection of hadronic events dominated by a single electromagnetic subcascade [48,49].

In the case of WCDs and SAs, the reconstruction of the shower parameters is obtained from the distribution of deposited energy or particle density at the ground level. The lateral distribution of particles in the shower is described by the Nishimura–Kamata–Greisen function [50]. Therefore, the reconstructed signals on the ground can be fit as a function of the distance from the core [42,51]. In HAWC, hadron-induced events are excluded based on the clumpiness of the lateral distribution (sporadic high signals at large distances from the core position, caused by muons and hadronic subshowers) [52]. Another parameter that is used is the deviation from the axial symmetry of the lateral distribution [42]. Optimal values of those cuts can be then selected, depending on the extent of the event in the detector (closely related to its energy). SAs with dedicated, underground detectors, such as Tibet AS-$\gamma$ and LHAASO-KM2A, can directly count the number of penetrative particles (mainly measuring muons) and exploit this information for gamma/hadron separation (see e.g., [53]). This allows for a background-free regime above about 100 TeV. Furthermore, the shower age parameter can be used to discriminate gamma rays from background events (see, e.g., [54]).

Machine learning methods using tree classifiers (such as random forest or boosted decision trees) are commonly used both in IACT [55–57] and WCD [58] data analysis to aggregate the information from different shower/image parameters into a single gamma/hadron separation parameter.

The method of arrival-direction reconstruction in WCDs is mainly based on the arrival time distribution along the instrument. This can be exploited with a planar fit [59]. Additional conical correction and iterative procedures can be used to further improve the direction reconstruction. For point-like sources, the angular resolution improvement also

lessens the effect of the isotropic background as it is integrated over a smaller solid angle around the position of the source.

Another method for the reconstruction of IACT events exploits the comparison of shower images with expected images either from a semi-analytic model [60,61] or using MC-generated templates [62]. Similarly, MC-template-based analysis has been also applied to WCD data [63]. Fitting of the individual events to such a model provides direct information about the physical properties of the shower (such as arrival direction, energy, impact, and height of the first interaction). The goodness of the fit can be used for gamma/hadron discrimination; however, this requires rather precise knowledge of the underlying probability distributions of the pixel response to gamma-ray showers (also including noise); hence, it is prone to systematic errors [64]. Hillas-based parameters in the IACT technique can also be combined with the parameters obtained from the shower model to obtain improved performance using multivariate methods [65].

*4.3. Background Modeling*

Even after gamma/hadron separation, TeV instruments still need to deal with the residual background[2]. It is essential for the background estimation to be determined precisely, because discrepancies of even a few percentage points can produce significant systematic errors in the measurement of source flux, possibly even causing an artificial signal. Due to this, for the definition of the sensitivity of IACTs, the expected excess must not only achieve $5\sigma$ statistical significance, but must also be above $\sim$5% of the residual background. To limit the systematic errors affecting the background estimation, the observation conditions, such as the zenith angle and atmosphere transparency, need to be as similar as possible for the "source" and "background" data. The simplest method to estimate the background for point-like sources is an analogue of aperture photometry. In the case of pointed instruments such as IACTs this can be realized by tracking a false source at a given offset from the actual target [66]. The source position will then slowly rotate around the camera center. Its reflected position (one or more) with respect to the center of the camera can be used as a background control region. To counteract the inhomogeneities of the camera, the positions of source and anti-source are swapped every few tens of minutes, in a process known as wobbling. Thanks to this method, the background can be estimated down to a systematic accuracy of $\sim$1% (see, e.g., [21]). Notably, such swapping of source and background positions will be only effective against instrumental inhomogeneities and not against those connected with the given sky position (such as stars or nearby, possibly extended, TeV sources). In particular, the presence of bright stars within $\sim$1° from the source position might require a dedicated correction procedure [67], such as simulating the effect of the star in the background.

With the total number of detected sources at TeV energies increasing, nearby TeV emitters are more and more prone to additionally complicate the background subtraction process. This is especially problematic in the case of crowded regions, such as the Galactic plane [22]. On one hand, such sources should be taken into account when background estimation regions are selected (see, e.g., [68]). In the case of scans with IACTs, as long as the inhomogeneity can be kept low, the background can be estimated from a ring around the source position, excluding parts of the ring that can be affected by other sources [22]; this is the so-called adaptive ring background method. On the other hand, the emissions of nearby sources (see, e.g., [69]) can overlap, either due to the actual coexistence in a particular direction of the sky of extended sources, or due to the smearing of the observed emission by the gamma-ray PSF of the instrument. Let us consider an example of two nearby sources, A and B. To obtain the flux of A, one needs to subtract the contribution of the B source, which depends on the flux of B. However, to obtain the flux of B, similarly one would need to know the flux of A. This can be solved using simultaneous likelihood fits of both sources with particular morphology and spectral models [70]. A similar method is commonly used in GeV satellite instruments (see e.g., [71]).

In the case of WCDs and SAs, different algorithms can be used to estimate the isotropic background. The equal-zenith method [13] is similar to aperture photometry, with the background taken from multiple OFF regions adjacent to the source. Due to the strong zenith angle dependence of the event rate, the background estimation regions are selected preferentially at the same instantaneous zenith angle as the source. The second method, direct integration algorithm [72], exploits the decomposition of the background into a time-independent angular distribution (expressed in local coordinates), as well as the direction-independence of the time-dependent total rate of events. This method is accurate down to a fraction of per mil, limited by small cosmic ray anisotropies [72]. Furthermore, in the case of WCDs, likelihood analysis has been used to fit the source morphology and take into account partially overlapping sources [73].

### 4.4. Largely Extended/Diffuse Emission

The spatial extension of TeV emissions can be challenging, in particular for IACT instruments. Firstly, because the underlying background is then larger, this considerably worsens the sensitivity of the instrument. Moreover, very extended sources, with a radius of $\gtrsim 1°$, have an extent comparable to that of IACT cameras. This results in a lack of background control regions, making analysis very difficult[3]. WCD and SA detectors, being wide-FOV instruments, are more suitable for such sources.

A specific case of extended sources is the all-sky diffuse emission. One example of this is the measurement of diffuse electron contribution to cosmic rays [75]. In such a case there is no possibility of estimating the background from the data themselves. Instead, MC simulations of protons are used and these are normalized to the data using proton-like events. In such studies, a careful selection of data quality and hadron rejection cuts is essential, as on one hand the background needs to be kept as low as possible to minimize systematic errors, but on the other hand, the studies are also sensitive to data/MC mismatches, which are often more pronounced for strong selection cuts. The diffuse emission of the background itself can be also studied to derive the cosmic ray spectrum [76,77]. Interestingly, this type of study is limited by the need for extensive MC generation rather than by the data collection process itself.

In contrast, the wide FOV of WCD/SA instruments, combined with the background direct integration method, allows the detection of sources with an extent of a few degrees [78]. WCDs and SAs are also excellent instruments for studying (diffuse) cosmic rays [79]. Large structures, such as the Galactic disk, can be efficiently studied with SA/WCD techniques as long as background control regions can be established [80]. Nevertheless, for all-sky fully diffuse emissions on top of the cosmic-ray background, they struggle with the same background estimation issue as IACTs.

### 4.5. Energy Spectrum

Due to both intrinsic shower fluctuations (see, e.g., [81]) and the measurement accuracy, there is considerable energy dispersion. This effect can cause the unbalanced migration of events between the considered energy bins, particularly for spectra that are either soft or have a cut-off. The conversion from estimated energies to true energies can be achieved by means of the unfolding procedure (see, e.g., [82] and references therein). Although the problem naively can be seen as a simple inversion of an energy migration matrix, the typical form of this matrix makes the inversion strongly dependent on even tiny changes of the input matrix. This can lead to nonphysical, oscillating solutions. There is no unique way to deal with this problem and different approaches might be more appropriate for individual cases. Most of the methods involve a free parameter (regularization) that describes the balance between the smoothness of the obtained solution, and its accuracy in the description of the data.

A special method of unfolding is the "forward folding" method. Specifically, a given source spectrum (expressed in true energy) is folded with the instrument response function into measurable quantities (typically the estimated energy). Comparison with the data can

be performed, e.g., by means of likelihood maximization. This method is in general more reliable than regular unfolding, in particular in the case of a strong energy migration. It can be also easily generalized to multiple or extended sources by performing a joint likelihood fit in terms of energy and direction. Both these reasons make it a natural method to be applied for wide-FOV instruments (see, e.g., [42] for application to WCDs); however, they have been also used for IACTs (see, e.g., [70]). A disadvantage of such a likelihood fit is the need for the assumption of a given source spectral form. Therefore, the result of the fit is only a set of spectral parameters, rather than spectral measurements at different energy ranges. Typically, simple phenomenological spectral shapes (power-law, log-parabola, power-law with a cut-off, etc.) are used. Although for a strong source complicated spectral shapes can be fit, it is usually only possible to fit sources of low significance with the simplest (i.e., having the least number of free parameters) spectral shape of the power-law. This neglects any intrinsic curvature of the spectra, and in some cases can bias the results derived from such studies, especially if they are made over a population of sources (specifically, although the curvature is not significant in an individual source, it is apparent in the combined analysis of multiple sources; see, e.g., discussion in [83]). It should be noted that even though the source spectrum at a given moment of time might closely follow a simple shape (e.g., a power-law), spectral variability can cause the effective spectrum integrated over an extended period of time to be quite complicated. Such complicated shapes cannot be parametrized easily, or the number of parameters can be too large for efficient fitting.

### 4.6. Deep Learning Methods

The standard analysis approach for both IACTs and WCDs involves the distillation of information from individual waveforms into information relating to pixels/tanks and further into a few parameters characterizing a given event. At each step, assumptions and simplifications are implemented that potentially causes losses of pieces of information and hence worsen the achieved performance of the method. An alternative, assumption-free approach is to provide as full information about each event as possible, and employ a machine learning method to find the best solution, using the so-called deep learning approach (DL, [84]). Such techniques can be used in all the stages of event reconstruction, for gamma/hadron separation, energy estimations or arrival direction estimations. Although this idea is not new (see, e.g., the application of neural network classifiers to Whipple telescope data, [85]) its popularity has risen rapidly in recent years, both due to the increase in the computer power and also due to development of publicly-available algorithms for the classification of optical images. A natural way of tackling the problem of the reconstruction of events based on its pixelized images is the use of convolutional neural networks (CNNs). This method exploits a sequential usage of filters for the detection of the image features and pooling layers that condense the information. On the other hand, recurrent neural networks (RNNs) are designed to handle data series with varying lengths, using the internal memory of the inputs to exploit the correlation between sequential inputs. Although their architecture is not directly suited for the analysis of two-dimensional events, their combination with CNNs has been shown to be an efficient way of combining information from multiple IACT telescopes [86,87].

There are a few challenges in the application of DL methods to TeV data. First, and most importantly, such methods are very sensitive to any differences between the two provided training sets. The reason for this is that the information about the shower is hidden in much more abundant pixels, tanks, or detector stations that carry only noise information. For example, if a gamma/hadron separation is trained on MC gamma-ray events and hadron events from real data, a DL method might be more prone to focus on the small MC/data discrepancies, rather than on the actual difference between gammas and hadrons [86]. Training on gamma-ray and hadron samples produced solely from MC simulations is possible but not always feasible. Specifically, hadronic showers are more CPU-time-consuming to generate and the samples are usually not complete

(both due to possibility of triggering small showers at large impact parameters[4], and due to the distribution of different nuclei present in the cosmic spectra). Additionally, contrary to the TeV gamma-ray MC simulations, which almost exclusively use a well-established electromagnetic process, the accuracy of cosmic-ray MC simulations suffers from the differences between available hadronic interaction models [88]. Moreover, when the training is performed using only MC simulations, its application to real data might show performance losses. On the contrary, training on data only is limited by the need for proper gamma-ray samples—even from the direction of strong sources, gamma rays are only a small fraction of the total number of events; hence, the preselection of particle type is needed. Furthermore, in this case, crucial information (such as the true energy of the primary particle) is not available.

Second, the efficient usage of DL methods requires large samples (the larger the sample, the more parameters are exploited), which in turn results in very long processing times. This has been mediated in recent years by the use of graphical processor units (see, e.g. [89]). Third, most of the generally available DL methods used for image analysis operate on square pixelization, which is common in regular images. In contrast, TeV instrumentation often uses a hexagonal pixel/tank configuration. Interpolation methods can be used to translate the events into square pixelization. However, dedicated DL methods using intrinsically hexagonal pixels are also available [90].

In the last few years, great progress in the application of DL methods to IACT instruments has been achieved. Although the first DL results performed with MC simulations in the past showed (sometimes large) improvements, when the training was applied to the actual data, the DL methods often turned out to perform worse. Nowadays there are multiple reports confirming improvements of DL-based analysis with real IACT data; however, the obtained improvement is still smaller than what is expected from MC-only studies. In the case of H.E.S.S. telescopes' data, DL methods were shown to provide a boost in the gamma/hadron separation of actual data; however, slight worsening in the angular resolution has been observed as well [86]. Similarly, in the case of LST-1 telescope commissioning data, DL methods provided an improvement in the gamma/hadron separation, but not in the angular resolution [91]. The CNN method applied to cleaned images of showers in MAGIC data resulted in similar sensitivity to the standard approach based on decision trees [92].

DL methods can go one step deeper and, in the case of pixel-wise information, use the whole, sampled PMT waveforms instead of single charge/time measurements per pixel [93]. An MC proof-of-concept study showed that this can further improve the gamma/hadron separation for IACT. However, this latest improvement still remains to be confirmed with actual IACT data.

Deep learning methods are also starting to be exploited for WCD and SA instruments. Although machine learning methods have been shown to improve the performance of gamma/hadron separation, they have been fed so far only with a small selection of event parameters, rather than the full event information [58]. The treating of HAWC events as images in CNN has been tried as well, and has shown a gamma/hadron separation capability [94]; however, it is not clear yet if such a method improves the performance. It is likely that as the understanding of these experiments improves in time, more complete information can be exploited as well. The use of deep learning for LHAASO-KM2A using MC simulations has shown an improvement in gamma/hadron separation with respect to the standard method based on the ratio of the detected muons (by underground detectors) and electrons/positrons (by surface detectors) [95]; however, the method has not yet been validated on the data.

### 4.7. Combination of Data from Different Instruments

Multiple TeV instruments are currently in operation. Often the information on the TeV emissions of a given source is available from more than one instrument, either by observing independently, responding to the same target of opportunity (TOO), or as a result of pre-

planned, joint observational campaigns. Although high-level products, such as light curves or spectra, may be sometimes combined using statistical methods, this is not always feasible due to, e.g., different time or energy binning used. Moreover, as the actual spectral points generally depend on the best fit of spectral shape, the inclusion of additional information from one experiment would change the spectrum obtained by the other. Finally, in the case of the non-detection of gamma-ray emissions, combining the upper limits from individual instruments is only possible if additional information (such as likelihood profiles; see, e.g., [96]) is available. Joint analysis of data from different instruments is marred by the fact that different (and sometimes proprietary) software and data formats are used by different collaborations. Nevertheless, in recent years a lot of progress has been achieved in terms of unifying the TeV astronomy data formats [97]. This allowed the first common analysis of data from different instruments performed within one framework [98]. Interestingly, that analysis involved not only WCD and IACT instruments, but also GeV satellite data. It should be noted, however, that for strong sources the statistical uncertainties of the individual measurements are generally much smaller than systematic differences between the instruments (see, e.g., [99]).

## 5. Multi-Wavelength and Multi-Messenger Observations

In order to better understand the processes governing cosmic sources, a broadband view is essential. The observations can be organized either as multi-wavelength (MWL), if the TeV band is combined with other electromagnetic observations (e.g., radio, optical, X-ray), or multi-messenger (MM) if gamma-ray observations are combined with studies of, e.g., neutrino emissions or gravitational waves. The common observations can be performed either in terms of pre-planned multi-instrument campaigns or as a response to a TOO.

In the case of TOO observations, WCDs and SAs, being wide-FOV instruments, have the advantage of being able to measure (if source is above the instrument's horizon) the source emission before and during the TOO time. However, the performance of those instruments over short time scales is relatively poor; hence, only extremely strong emissions can be probed in this way. On the other hand, IACTs, if designed for this goal and equipped with appropriate software for automatic reaction, are able to repoint to any position in the visible sky within a timescale of a few tens of seconds (see, e.g., [11]). It should be noted, however, that such immediate IACT observations might encounter non-optimal conditions (such as high zenith distance angle, bad weather conditions, or moonlight, all of which increase the energy threshold of the observations).

*TOO Observations with a Large Position Uncertainty*

A particularly difficult case is the follow-up observations of single events which cannot be associated firmly to a known source and thus have a large position uncertainty. The positional accuracy of MWL triggers, such as fast radio bursts [100] or gamma-rays bursts [101], normally depends on the electromagnetic flux of the event and the instrument from which the alert originated. The uncertainty of the position thus typically varies between much less than the IACT PSF and a fraction of a degree.

The localiation accuracy is usually worse in the follow-up of MM triggers. In particular, the current localization accuracy of the gravitational wave events [102] can easily cover a region of tens of square degrees in the sky, which is significantly larger than the FOV of IACTs. This requires a clever pointing strategy that maximizes the chance of detection of the electromagnetic counterpart (see, e.g., [103]). In the case of a follow-up of a single high-energy neutrino event [104], the localization accuracy is usually much better (of the order of a fraction of a degree), which is, however, still considerably larger than the PSF of IACTs.

This has a direct effect on the detection capability of such events. For a source with an unknown location, it is not sufficient to achieve the detection of an excess from some location within the error circle of the neutrino localization with a $5\sigma$ statistical significance,

because the number of trials associated with the different possible locations of the source has to be taken into account. As an example, if the localization uncertainty is ~1° and the PSF is ~0.1°, the number of trials can be roughly estimated as ~100. In such a situation, in order to achieve a detection at the $5\sigma$ level after trials, at least a $5.8\sigma$ pre-trial excess is needed in one of the investigated locations within the location uncertainty of the event. This example effectively corresponds to a ~ 15% worsening of the sensitivity. Additionally, a possible drop in acceptance at higher offsets from the IACT camera center can further worsen the performance.

The situation is even more complicated if no significant emission is detected and if one is interested in putting constraints on the emission of the event with an uncertain location. The upper limit of the emission exploits the observed signal [105]; specifically, a given flux value can be consistent with a small positive excess, but would be unlikely if a null or negative excess is observed. In the sky region covering the localization error of the event, even if no true emission is present, it is common to observe excesses with a statistical significance of 2–3$\sigma$, merely due to the combination of the Gaussian distribution of the used test statistic and the number of trials. The limit of the emission in those particular sky positions will be then larger, even by a factor of a few, than in the average location on the sky. As one of those positions might in fact hide a real, weak source, the most conservative approach for putting a limit on the TeV emission associated with the poorly localized event would be the least constraining one, which severely worsens the performance for constraining emission models.

WCDs and SAs, being wide-FOV instruments, are more suitable for the follow-up of poorly localized alerts as they can simultaneously scan a large area in the sky. Although they are also affected by the problem of the number of trials in the search for such an emission, as their angular resolution is usually worse than of IACTs, the corresponding effect of trials is lesser. However, for the time scale of transient sources, the sensitivity of WCDs/SAs are usually significantly worse than that of IACTs (see, e.g., [106]).

## 6. Conclusions

TeV astronomy is undergoing its golden age. Most of the TeV emissions of the ~250 sources currently known in this energy range were discovered using the current generation of instruments. Moreover, they cover a range of various types of objects, showing the versatility of the IACT, WCD, and SA techniques. It is to be expected that those sources are only the tip of the iceberg of the sources available to the upcoming generation of instruments. Although the standard analysis techniques for both IACT and WCD/SA instruments are well established, there is still an ongoing effort aimed at improving them. Recently, this effort has concentrated on one hand on the usage of the DL techniques. On the other hand, there is also a growing interest in the development of open tools and common data formats that facilitate the analysis and allow the simple combination of data sets from various experiments.

Although IACT, WCD, and SA instruments operate in a similar energy range, the differences of the used techniques cause differences in their operation modes and performance parameters, which allows them to complement each other. The use of all these types of techniques is also desirable in joint MWL or MM studies with other instruments.

**Funding:** This research was funded by Narodowe Centrum Nauki grant number 2019/34/E/ST9/ 00224.

**Acknowledgments:** The author would like to thank Takashi Sako, Rubén López-Coto, and Harm Schoorlemmer, as well as anonymous journal reviewers for providing useful comments on the manuscript.

**Conflicts of Interest:** The author declares no conflict of interest.

## Abbreviations

The following abbreviations are used in this manuscript:

| | |
|---|---|
| CNN | Convolutional Neural Networks |
| CTA | Cherenkov Telescope Array |
| DL | Deep Learning |
| FOV | Field of View |
| HAWC | High-Altitude Water Cherenkov |
| H.E.S.S. | High-Energy Stereoscopic System |
| IACT | Imaging Atmospheric Cherenkov Telescope |
| LHAASO | Large High-Altitude Air Shower Observatory |
| LHAASO-WCDA | Water Cherenkov Detector Array |
| LHAASO-WFCTA | Wide-Field Air Cherenkov Telescope Array |
| LST | Large-Sized Telescope |
| MAGIC | Major Atmospheric Gamma-Ray Imaging Cherenkov |
| MC | Monte Carlo (Simulations) |
| MM | Multi-Messenger |
| MWL | Multi-Wavelength |
| PMT | Photomultiplier Tube |
| PSF | Point Spread Function |
| RNN | Recurrent Neural Networks |
| SA | Surface Array |
| SiPM | Silicon Photomultiplier |
| SWGO | Southern Wide-field Gamma-Ray Observatory |
| TOO | Target of Opportunity |
| VERITAS | Very Energetic Radiation Imaging Telescope Array System |
| VHE | Very High Energy |
| WCD | Water Cherenkov Detector |

## Notes

[1] Various names are used for those types of detectors, including surface arrays, surface detectors, air shower arrays

[2] Only at the highest energies for strong sources can the observations be considered background-free.

[3] If the extension of the region is mainly in one direction, such as the inner part of the Galactic plane [74], it is still possible to use the adaptive ring method to evaluate the background.

[4] In the case of IACTs also at large offset angles from the camera center

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
