# Peer review of "TeV Instrumentation: Current and Future"

_galaxies, doi:10.3390/galaxies10010021_

Round 1

Reviewer 1 Report

This nice review summarizes the current status of the TeV astronomy instrumentation, primarily concentrating on the comparison of various types of equipment and analysis and providing an insight into future installations (e.g. CTA, SWGO, ALPACA).

Strength points: 

Explaining the aim of the article was vivid. 

Containing a coherent introduction.

Containing an extensive review of techniques.

Referring to the observational data sources and several experiments.

Author Response

Thank you for the reading of the manuscript and for positive review.

Reviewer 2 Report

Attached file

Author Response

Thank you for your careful review. The manuscript have been updated accordingly and in the attached .pdf file I include responses point by point.

Reviewer 3 Report

The manuscript reviews the current status of the TeV gamma-ray astronomy instrumentation. Author compare different techniques of observations of gamma-ray sources, discussing their limitations and advantages. Also discussed is data analysis, and in particular application of deep learning technique, which is ubiquitous in many branches of science and studying its applications is a novel, hot topic. The manuscript is written clearly (reading is very smooth) and it references relevant papers.

I have noticed two typos: line 595, should be “with an unknown”, line 598 “uncertainty is”

Author Response

Thank you for the reading of the manuscript. Both typos are corrected now.

Round 2

Reviewer 2 Report

The paper has been improved by responding to the main requests of the referee. It is therefore worth of publication in this form

a couple of typos
line 112 "producion" should be "production"
line 336 "LHASSO-KM2A" should be "LHAASO-KM2A"

Author Response

Thank you for checking again the manuscript. Both typos are now corrected.